# Software citation principles

Arfon M. Smith[1,*], Daniel S. Katz[2,*], Kyle E. Niemeyer[3,*] and
FORCE11 Software Citation Working Group

[1] GitHub, Inc., San Francisco, California, United States
[2] National Center for Supercomputing Applications & Electrical and Computer Engineering
Department & School of Information Sciences, University of Illinois at Urbana-Champaign,
Urbana, Illinois, United States
[3] School of Mechanical, Industrial, and Manufacturing Engineering, Oregon State University,
Corvallis, Oregon, United States
* These authors contributed equally to this work.

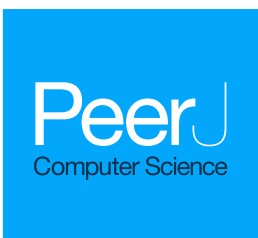

## ABSTRACT

Software is a critical part of modern research and yet there is little support across the
scholarly ecosystem for its acknowledgement and citation. Inspired by the activities
of the FORCE11 working group focused on data citation, this document
summarizes the recommendations of the FORCE11 Software Citation Working
Group and its activities between June 2015 and April 2016. Based on a review of
existing community practices, the goal of the working group was to produce a
consolidated set of citation principles that may encourage broad adoption of a
consistent policy for software citation across disciplines and venues. Our work is
presented here as a set of software citation principles, a discussion of the motivations
for developing the principles, reviews of existing community practice, and a
discussion of the requirements these principles would place upon different
stakeholders. Working examples and possible technical solutions for how these
principles can be implemented will be discussed in a separate paper.

## SOFTWARE CITATION PRINCIPLES

The main contribution of this document are the software citation principles, written fairly
concisely in this section and discussed further later in the document (see Discussion).
In addition, we also motivate the creation of these principles (see Motivation),
describe the process by which they were created (see Process of Creating Principles),
summarize use cases related to software citation (see Use Cases), and review related
work (see Related Work). We also lay out the work needed to lead to these software
citation principles being applied (see Future Work).

1. **Importance:** Software should be considered a legitimate and citable product of
   research. Software citations should be accorded the same importance in the scholarly
   record as citations of other research products, such as publications and data; they
   should be included in the metadata of the citing work, for example in the reference list
   of a journal article, and should not be omitted or separated. Software should be cited on
   the same basis as any other research product such as a paper or a book, that is, authors
   should cite the appropriate set of software products just as they cite the appropriate set
   of papers.

Corresponding author
Daniel S. Katz, d.katz@ieee.org

2. **Credit and attribution:** Software citations should facilitate giving scholarly credit and normative, legal attribution to all contributors to the software, recognizing that a single style or mechanism of attribution may not be applicable to all software.

3. **Unique identification:** A software citation should include a method for identification that is machine actionable, globally unique, interoperable, and recognized by at least a community of the corresponding domain experts, and preferably by general public researchers.

4. **Persistence:** Unique identifiers and metadata describing the software and its disposition should persist—even beyond the lifespan of the software they describe.

5. **Accessibility:** Software citations should facilitate access to the software itself and to its associated metadata, documentation, data, and other materials necessary for both humans and machines to make informed use of the referenced software.

6. **Specificity:** Software citations should facilitate identification of, and access to, the specific version of software that was used. Software identification should be as specific as necessary, such as using version numbers, revision numbers, or variants such as platforms.

## MOTIVATION

As the process of research[1] has become increasingly digital, research outputs and products have grown beyond simply papers and books to include software, data, and other electronic components such as presentation slides, posters, (interactive) graphs, maps, websites (e.g., blogs and forums), and multimedia (e.g., audio and video lectures). Research knowledge is embedded in these components. Papers and books themselves are also becoming increasingly digital, allowing them to become executable and reproducible. As we move towards this future where research is performed in and recorded as a variety of linked digital products, the characteristics and properties that developed for books and papers need to be applied to, and possibly adjusted for, all digital products. Here, we are concerned specifically with the citation of software products. The challenge is not just the textual citation of software in a paper, but the more general identification of software used within the research process. This work focuses on making software a citable entity in the scholarly ecosystem. While software products represent a small fraction of the sum total of research output, this work together with other efforts such as the FORCE11 Data Citation Principles (*Data Citation Synthesis Group, 2014*; *Starr et al., 2015*) collectively represent an effort to better describe (and cite) all outputs of research.

Software and other digital resources currently appear in publications in very inconsistent ways. For example, a random sample of 90 articles in the biology literature found seven different ways that software was mentioned, including simple names in the full-text, URLs in footnotes, and different kinds of mentions in reference lists: project names or websites, user manuals or publications that describe or introduce the software (*Howison & Bullard, 2015*). Table 1 shows examples of these varied forms of software mentions and the frequency with which they were encountered. Many of these kinds of mentions fail to perform the functions needed

[1] We use the term "research" in this document to include work intended to increase human knowledge and benefit society, in science, engineering, humanities, and other areas.

**Table 1 Varieties of software mentions in publications, from *Howison & Bullard (2015)*.**

| Mention type | Count (n = 286) | Percentage (%) |
|---|---|---|
| Cite to publication | 105 | 37 |
| Cite to user's manual | 6 | 2 |
| Cite to name or website | 15 | 5 |
| Instrument-like | 53 | 19 |
| URL in text | 13 | 5 |
| In-text name only | 90 | 31 |
| Not even name | 4 | 1 |

of citations, and their very diversity and frequent informality undermine the integration of software work into bibliometrics and other analyses. Studies on data and facility citation have shown similar results (*Huang, Rose & Hsu, 2015*; *Mayernik, Maull & Hart, 2015*; *Parsons, Duerr & Minster, 2010*).

There are many reasons why this lack of both software citations in general and standard practices for software citation are of concern:

- Understanding research fields: Software is a product of research, and by not citing it we leave holes in the record of research of progress in those fields.
- Credit: Academic researchers at all levels, including students, postdocs, faculty, and staff, should be credited for the software products they develop and contribute to, particularly when those products enable or further research done by others.[2] Non-academic researchers should also be credited for their software work, though the specific forms of credit are different than for academic researchers.
- Discovering software: Citations enable the specific software used in a research product to be found. Additional researchers can then use the same software for different purposes, leading to credit for those responsible for the software.
- Reproducibility: Citation of specific software used is necessary for reproducibility, although not sufficient. Additional information such as configurations and platform issues are also needed.

[2] Providing recognition of software can have tremendous economic impact as demonstrated by the role of Text REtrieval Conference (TREC) in information retrieval (*Rowe et al., 2010*).

## PROCESS OF CREATING PRINCIPLES

The FORCE11 Software Citation Working Group was created in April 2015 with the following mission statement:

> *The software citation working group is a cross-team committee leveraging the perspectives from a variety of existing initiatives working on software citation to produce a consolidated set of citation principles in order to encourage broad adoption of a consistent policy for software citation across disciplines and venues. The working group will review existing efforts and make a set of recommendations. These recommendations will be put off for endorsement by the organizations represented by this group and others that play an important role in the community.*

*The group will produce a set of principles, illustrated with working examples, and a plan for dissemination and distribution. This group will not be producing detailed specifications for implementation although it may review and discuss possible technical solutions.*

The group gathered members (see Appendix A) in April and May 2015, and then began work in June. This materialized as a number of meetings and offline work by group members to document existing practices in member disciplines; gather materials from workshops and other reports; review those materials, identifying overlaps and differences; create a list of use cases related to software citation, recorded in Appendix B; and subsequently draft an initial version of this document. The draft Software Citation Principles document was discussed in a day-long workshop and presented at the FORCE2016 Conference in April 2016 (https://www.force11.org/meetings/force2016). Members of the workshop and greater FORCE11 community gave feedback, which we recorded here in Appendix C. This discussion led to some changes in the use cases and discussion, although the principles themselves were not modified. We also plan to initiate a follow-on implementation working group that will work with stakeholders to ensure that these principles impact the research process.

The process of creating the software citation principles began by adapting the FORCE11 Data Citation Principles (*Data Citation Synthesis Group, 2014*). These were then modified based on discussions of the FORCE11 Software Citation Working Group (see Appendix A for members), information from the use cases in section Use Cases, and the related work in section Related Work.

We made the adaptations because software, while similar to data in terms of not traditionally having been cited in publications, is also different than data. In the context of research (e.g., in science), the term "data" usually refers to electronic records of observations made in the course of a research study ("raw data") or to information derived from such observations by some form of processing ("processed data"), as well as the output of simulation or modeling software ("simulated data"). Some confusion about the distinction between software and data comes in part from the much wider scope of the term "data" in computing and information science, where it refers to anything that can be processed by a computer. In that sense, software is just a special kind of data. Because of this, citing software is not the same as citing data. A more general discussion about these distinctions is currently underway (https://github.com/danielskatz/software-vs-data).

The principles in this document should guide further development of software citation mechanisms and systems, and the reader should be able to look at any particular example of software citation to see if it meets the principles. While we strive to offer practical guidelines that acknowledge the current incentive system of academic citation, a more modern system of assigning credit is sorely needed. It is not that academic software needs a separate credit system from that of academic papers, but that the need for credit for research software underscores the need to overhaul the system of credit for all research products. One possible solution for a more complete

description of the citations and associated credit is the transitive credit proposed by *Katz (2014)* and *Katz & Smith (2015)*.

## USE CASES

We documented and analyzed a set of use cases related to software citation in FORCE11 Software Citation Working Group (https://docs.google.com/document/d/1dS0SqGoBIFwLB5G3HiLLEOSAAgMdo8QPEpjYUaWCvIU) (recorded in Appendix B for completeness). Table 2 summarizes these use cases and makes clear what the requirements are for software citation in each case. Each example represents a particular stakeholder performing an activity related to citing software, with the given metadata as information needed to do that. In that table, we use the following definitions:

- "Researcher" includes both academic researchers (e.g., postdoc, tenure-track faculty member) and research software engineers.
- "Publisher" includes both traditional publishers that publish text and/or software papers as well as archives such as Zenodo that directly publish software.
- "Funder" is a group that funds software or research using software.
- "Indexer" examples include Scopus, Web of Science, Google Scholar, and Microsoft Academic Search.
- "Domain group/library/archive" includes the Astronomy Source Code Library (ASCL; http://ascl.net); biomedical and healthCAre Data Discovery Index Ecosystem (bioCADDIE; https://biocaddie.org); Computational Infrastructure for Geodynamics (CIG; https://geodynamics.org), libraries, institutional archives, etc.
- "Repository" refers to public software repositories such as GitHub, Netlib, Comprehensive R Archive Network (CRAN), and institutional repositories.
- "Unique identifier" refers to unique, persistent, and machine-actionable identifiers such as a DOI, ARK, or PURL.
- "Description" refers to some description of the software such as an abstract, README, or other text description.
- "Keywords" refers to keywords or tags used to categorize the software.
- "Reproduce" can mean actions focused on reproduction, replication, verification, validation, repeatability, and/or utility.
- "Citation manager" refers to people and organizations that create scholarly reference management software and websites including Zotero, Mendeley, EndNote, RefWorks, BibDesk, etc., that manage citation information and semi-automatically insert those citations into research products.

All use cases assume the existence of a citable software object, typically created by the authors/developers of the software. Developers can achieve this by, e.g., uploading a software release to figshare (https://figshare.com/) or Zenodo (*GitHub, 2014*) to obtain a DOI. Necessary metadata should then be included in a `CITATION` file (*Wilson, 2013*) or machine-readable `CITATION.jsonld` file (*Katz & Smith, 2015*). When software is not

**Table 2 Use cases and basic metadata requirements for software citation, adapted from FORCE11 Software Citation Working Group.** Solid circles (•) indicate that the use case depends on that metadata, while plus signs (+) indicate that the use case would benefit from that metadata if available.

| Use case | Basic requirements | | | | | | | | | | | Example stakeholder(s) |
|---|---|---|---|---|---|---|---|---|---|---|---|---|
| | Unique identifier | Software name | Author(s) | Contributor role | Version number | Release date | Location/ repository | Indexed citations | Software license | Description | Keywords | |
| 1. Use software for a paper | • | • | • | | • | • | • | | + | + | | Researcher |
| 2. Use software in/with new software | • | • | • | | • | • | • | | + | + | | Researcher, software engineer |
| 3. Contribute to software | • | • | • | + | • | • | • | | + | + | | Researcher, software engineer |
| 4. Determine use/citations of software | • | • | • | | | | | • | | | | Researcher, software engineer |
| 5. Get credit for software development | • | • | • | + | • | • | • | + | | | | Researcher |
| 6. "Reproduce" analysis | • | • | • | | • | • | • | | + | + | | Researcher |
| 7. Find software to implement task | • | • | • | | | • | • | • | + | + | + | Researcher, software engineer |
| 8. Publish software paper | • | • | • | | • | • | • | • | | | | Publisher |
| 9. Publish papers that cite software | • | • | • | | • | • | • | • | | | | Publisher |
| 10. Build catalog of software | • | • | • | | • | • | • | • | + | + | + | Indexer |
| 11. Build software catalog/registry | • | • | • | | | | • | | | + | + | Domain group, library, archive |
| 12. Show scientific impact of holdings | • | • | • | | • | • | | • | | | | Repository |
| 13. Show how funded software has been used | • | • | • | | | | | • | | | | Funder, policy maker |
| 14. Evaluate contributions of researcher | • | • | • | + | | • | | • | | | | Evaluator, funder |
| 15. Store software entry | • | • | • | | • | • | • | | | + | + | Citation manager |
| 16. Publish mixed data/software packages | • | • | • | | • | • | • | | + | + | + | Repository, library, archive |

freely available (e.g., commercial software) or when there is no clear identifier to use, alternative means may be used to create citable objects as discussed in section Access to Software.

In some cases, if particular metadata are not available, alternatives may be provided. For example, if the version number and release date are not available, the download date can be used. Similarly, the contact name/email is an alternative to the location/repository.

## RELATED WORK

With approximately 50 working group participants (see Appendix A) representing a range of research domains, the working group was tasked to document existing practices in their respective communities. A total of 47 documents were submitted by working group participants, with the life sciences, astrophysics, and geosciences being particularly well-represented in the submitted resources.

### General community/non domain-specific activities

Some of the most actionable work has come from the UK Software Sustainability Institute (SSI) in the form of blog posts written by their community fellows. For example, in a blog post from 2012, *Jackson (2012)* discusses some of the pitfalls of trying to cite software in publications. He includes useful guidance for when to consider citing software as well as some ways to help "convince" journal editors to allow the inclusion of software citations.

*Wilson (2013)* suggests that software authors include a `CITATION` file that documents exactly how the authors of the software would like to be cited by others. While this is not a formal metadata specification (e.g., it is not machine readable) this does offer a solution for authors wishing to give explicit instructions to potential citing authors and, as noted in the motivation section (see Motivation), there is evidence that authors follow instructions if they exist (*Huang, Rose & Hsu, 2015*).

In a later post on the SSI blog, Jackson gives a good overview of some of the approaches package authors have taken to automate the generation of citation entities such as BibTeX entries (*Jackson, 2014*), and *Knepley et al. (2013)* do similarly.

While not usually expressed as software citation principles, a number of groups have developed community guidelines around software and data citation. *Van de Sompel et al. (2004)* argue for registration of all units of scholarly communication, including software. In "Publish or be damned? An alternative impact manifesto for research software," *Chue Hong (2011)* lists nine principles as part of "The Research Software Impact Manifesto." In the "Science Code Manifesto" (*Barnes et al., 2016*), the founding signatories cite five core principles (Code, Copyright, Citation, Credit, Curation) for scientific software.

Perhaps in light of the broad range of research domains struggling with the challenge of better recognizing the role of software, funders and agencies in both the US (e.g., NSF, NIH, Alfred P. Sloan Foundation) and UK (e.g., SFTC, JISC, Wellcome Trust) have sponsored or hosted a number of workshops with participants from across a range of

disciplines, specifically aimed at discussing issues around software citation (*Sufi et al., 2014*; *Ahalt et al., 2015*; *Software Credit Workshop, 2015*; *Norén, 2015*; *Software Attribution for Geoscience Applications, 2015*; *Allen et al., 2015*). In many cases these workshops produced strong recommendations for their respective communities on how best to proceed. In addition, a number of common themes arose in these workshops, including (1) the critical need for making software more "citable" (and therefore actions authors and publishers should take to improve the status quo), (2) how to better measure the impact of software (and therefore attract appropriate funding), and (3) how to properly archive software (where, how, and how often) and how this affects what to cite and when.

Most notable of the community efforts are those of WSSSPE Workshops (http://wssspe.researchcomputing.org.uk/) and SSI Workshops (http://www.software.ac.uk/community/workshops), who between them have run a series of workshops aimed at gathering together community members with an interest in (1) defining the set of problems related to the role of software and associated people in research settings, particularly academia, (2) discussing potential solutions to those problems, (3) beginning to work on implementing some of those solutions. In each of the three years that WSSSPE workshops have run thus far, the participants have produced a report (*Katz et al., 2014*; *Katz et al., 2016a*; *Katz et al., 2016b*) documenting the topics covered. Section 5.8 and Appendix J in the WSSSPE3 report (*Katz et al., 2016b*) has some preliminary work and discussion particularly relevant to this working group. In addition, a number of academic publishers such as APA (*McAdoo, 2015*) have recommendations for submitting authors on how to cite software, and journals such as *F1000Research* (http://f1000research.com/for-authors/article-guidelines/software-tool-articles), *SoftwareX* (http://www.journals.elsevier.com/softwarex/), *Open Research Computation* (http://www.openresearchcomputation.com) and the *Journal of Open Research Software* (http://openresearchsoftware.metajnl.com) allow for submissions entirely focused on research software.

### Domain-specific community activities

One approach to increasing software "citability" is to encourage the submission of papers in standard journals describing a piece of research software, often known as software papers (see Software Papers). While some journals (e.g., Transactions on Mathematical Software (TOMS), Bioinformatics, Computer Physics Communications, F1000Research, Seismological Research Letters, Electronic Seismologist) have traditionally accepted software submissions, the American Astronomical Society (AAS) has recently announced they will accept software papers in their journals (*AAS Editorial Board, 2016*). Professional societies are in a good position to change their respective communities, as the publishers of journals and conveners of domain-specific conferences; as publishers they can change editorial policies (as AAS has done) and conferences are an opportunity to communicate and discuss these changes with their communities.

In astronomy and astrophysics: The Astrophysics Source Code Library (ASCL; http://ASCL.net) is a website dedicated to the curation and indexing of software used in the astronomy-based literature. In 2015, the AAS and GitHub co-hosted a workshop

(*Norén, 2015*) dedicated to software citation, indexing, and discoverability in astrophysics. More recently, a Birds of a Feather session was held at the Astronomical Data Analysis Software and Systems (ADASS) XXV conference (*Allen et al., 2015*) that included discussion of software citation.

In the life sciences: In May 2014, the NIH held a workshop aimed at helping the biomedical community discover, cite, and reuse software written by their peers. The primary outcome of this workshop was the Software Discovery Index Meeting Report (*White et al., 2014*) which was shared with the community for public comment and feedback. The authors of the report discuss what framework would be required for supporting a Software Discovery Index including the need for unique identifiers, how citations to these would be handled by publishers, and the critical need for metadata to describe software packages.

In the geosciences: The Ontosoft (*Gil, Ratnakar & Garijo, 2015*) project describes itself as "A Community Software Commons for the Geosciences." Much attention was given to the metadata required to describe, discover, and execute research software. The NSF-sponsored Geo-Data Workshop 2011 (*Fox & Signell, 2011*) revolved around data lifecycle, management, and citation. The workshop report includes many recommendations for data citation.

## Existing efforts around metadata standards

Producing detailed specifications and recommendations for possible metadata standards to support software citation was not within the scope of this working group. However some discussion on the topic did occur and there was significant interest in the wider community to produce standards for describing research software metadata.

Content specifications for software metadata vary across communities, and include DOAP (https://github.com/edumbill/doap/), an early metadata term set used by the Open Source Community, as well as more recent community efforts like Research Objects (*Bechhofer et al., 2013*), The Software Ontology (*Malone et al., 2014*), EDAM Ontology (*Ison et al., 2013*), Project CRediT (*CRediT, 2016*), the OpenRIF Contribution Role Ontology (*Gutzman et al., 2016*), Ontosoft (*Gil, Ratnakar & Garijo, 2015*), RRR/JISC guidelines (*Gent, Jones & Matthews, 2015*), or the terms and classes defined at schema.org related to the https://schema.org/SoftwareApplication class. In addition, language-specific software metadata schemes are in widespread use, including the Debian package format (*Jackson & Schwarz, 2016*), Python package descriptions (*Ward & Baxter, 2016*), and R package descriptions (*Wickham, 2015*), but these are typically conceived for software build, packaging, and distribution rather than citation. CodeMeta (*Jones et al., 2014*) has created a crosswalk among these software metadata schemes and an exchange format that allows software repositories to effectively interoperate.

## DISCUSSION

In this section we discuss some the issues and concerns related to the principles stated in section Software Citation Principles.

## What software to cite

The software citation principles do not define what software should be cited, but rather how software should be cited. What software should be cited is the decision of the author(s) of the research work in the context of community norms and practices, and in most research communities, these are currently in flux. In general, *we believe that software should be cited on the same basis as any other research product such as a paper or book*; that is, authors should cite the appropriate set of software products just as they cite the appropriate set of papers, perhaps following the FORCE11 Data Citation Working Group principles, which state, "In scholarly literature, whenever and wherever a claim relies upon data, the corresponding data should be cited" (*Data Citation Synthesis Group, 2014*).

Some software which is, or could be, captured as part of data provenance may not be cited. Citation is partly a record of software important to a research outcome[3], where provenance is a record of all steps (including software) used to generated particular data within the research process. Research results, including data, increasingly depend on software (*Hannay et al., 2009*), and thus may depend on the specific version used (*Sandve et al., 2013*; *Wilson et al., 2014*). Furthermore, errors in software or environment variations can affect results (*Morin et al., 2012*; *Soergel, 2015*). This implies that for a data research product, provenance data will include some of the cited software. Similarly, the software metadata recorded as part of data provenance will overlap the metadata recorded as part of software citation for the software that was used in the work. The data recorded for reproducibility should also overlap the metadata recorded as part of software citation. In general, we intend the software citation principles to cover the minimum of what is necessary for software citation for the purpose of software identification. Some use cases related to citation (e.g., provenance, reproducibility) might have additional requirements beyond the basic metadata needed for citation, as Table 2 shows.

## Software papers

Currently, and for the foreseeable future, software papers are being published and cited, in addition to software itself being published and cited, as many community norms and practices are oriented towards citation of papers. As discussed in the Importance principle (1) and the discussion above, *the software itself should be cited on the same basis as any other research product; authors should cite the appropriate set of software products.* If a software paper exists and it contains results (performance, validation, etc.) that are important to the work, then the software paper should also be cited. We believe that a request from the software authors to cite a paper should typically be respected, and the paper cited *in addition to* the software.

## Derived software

The goals of software citation include the linked ideas of crediting those responsible for software and understanding the dependencies of research products on specific software. In the Importance principle (1), we state that "software should be cited on the same basis as any other research product such as a paper or a book; that is, authors should cite

[3] Citation can be used for many purposes, including for software: which software has been used in the work, which software has influenced the work, which software is the work superseding, which software is the work disproving, etc.

the appropriate set of software products just as they cite the appropriate set of papers." In the case of one code that is derived from another code, citing the derived software may appear to not credit those responsible for the original software, nor recognize its role in the work that used the derived software. However, this is really analogous to how any research builds on other research, where each research product just cites those products that it directly builds on, not those that it indirectly builds on. Understanding these chains of knowledge and credit have been part of the history of science field for some time, though more recent work suggests more nuanced evaluation of the credit chains (*CRediT, 2016*; *Katz & Smith, 2015*).

## Software peer review

Adherence to the software citation principles enables better peer review through improved reproducibility. However, since the primary goal of software citation is to identify the software that has been used in a scholarly product, the peer review of software itself is mostly out of scope in the context of software citation principles. For instance, when identifying a particular software artifact that has been used in a scholarly product, whether or not that software has been peer-reviewed is irrelevant. One possible exception would be if the peer-review status of the software should be part of the metadata, but the working group does not believe this to be part of the minimal metadata needed to identify the software.

## Citation format in reference list

Citations in references in the scholarly literature are formatted according to the citation style (e.g., AMS, APA, Chicago, MLA) used by that publication. (Examples illustrating these styles have been published by *Lipson (2011)*; the follow-on Software Citation Implementation Group will provide suggested examples.) As these citations are typically sent to publishers as text formatted in that citation style, not as structured metadata, and because the citation style dictates how the human reader sees the software citation, *we recommend that all text citation styles support the following: a) a label indicating that this is software, e.g., [Software], potentially with more information such as [Software: Source Code], [Software: Executable], or [Software: Container], and b) support for version information, e.g., Version 1.8.7.*

## Citations limits

This set of software citation principles, if followed, will cause the number of software citations in scholarly products to increase, thus causing the number of overall citations to increase. Some scholarly products, such as journal articles, may have strict limits on the number of citations they permit, or page limits that include reference sections. Such limits are counter to our recommendation, and *we recommend that publishers using strict limits for the number of citations add specific instructions regarding software citations to their author guidelines to not disincentivize software citation.* Similarly, *publishers should not include references in the content counted against page limits.*

## Unique identification

The Unique Identification principle (3) calls for "a method for identification that is machine actionable, globally unique, interoperable, and recognized by a community." What this means for data is discussed in detail in the "Unique Identification" section of a report by the FORCE11 Data Citation Implementation Group (*Starr et al., 2015*), which calls for "unique identification in a manner that is machine-resolvable on the Web and demonstrates a long-term commitment to persistence." This report also lists examples of identifiers that match these criteria including DOIs, PURLs, Handles, ARKS, and NBNs. For software, *we recommend the use of DOIs as the unique identifier due to their common usage and acceptance, particularly as they are the standard for other digital products such as publications.*

While we believe there is value in including the explicit version (e.g., Git SHA1 hash, Subversion revision number) of the software in any software citation, there are a number of reasons that a commit reference together with a repository URL is not recommended for the purposes of software citation:

1. Version numbers/commit references are not guaranteed to be permanent. Projects can be migrated to new version control systems (e.g., SVN to Git). In addition, it is possible to overwrite/clobber a particular version (e.g., force-pushing in the case of Git).
2. A repository address and version number does not guarantee that the software is available at a particular (resolvable) URL, especially as it is possible for authors to remove their content from, e.g., GitHub.
3. A particular version number/commit reference may not represent a "preferred" point at which to cite the software from the perspective of the package authors.

We recognize that there are certain situations where it may not be possible to follow the recommended best-practice. For example, if (1) the software authors did not register a DOI and/or release a specific version, or (2) the version of the software used does not match what is available to cite. In those cases, falling back on a combination of the repository URL and version number/commit hash would be an appropriate way to cite the software used.

Note that the "unique" in a UID means that it points to a unique, specific software version. However, multiple UIDs might point to the same software. This is not recommended, but is possible. *We strongly recommend that if there is already a UID for a version of software, no additional UID should be created.* Multiple UIDs can lead to split credit, which goes against the Credit and Attribution principle (2).

### Software versions and identifiers

There are at least three different potential relationships between identifiers and versions of software:

1. An identifier can point to a specific version of a piece of software.
2. An identifier can point to the piece of software, effectively all versions of the software.
3. An identifier can point to the latest version of a piece of software.

It is possible that a given piece of software may have identifiers of all three types. In addition, there may be one or more software papers, each with an identifier.

While we often need to cite a specific version of software, we may also need a way to cite the software in general and to link multiple releases together, perhaps for the purpose of understanding citations to the software. The principles in section Software Citation Principles are intended to be applicable at all levels, and to all types of identifiers, such as DOIs, RRIDs, etc., though we again recommend when possible the use of DOIs that identify specific versions of source code. We note that RRIDs were developed by the FORCE11 Resource Identification Initiative (https://www.force11.org/group/resource-identification-initiative) and have been discussed for use to identify software packages (not specific versions), though the FORCE11 Resource Identification Technical Specifications Working Group (https://www.force11.org/group/resource-identification-technical-specifications-working-group) says "Information resources like software are better suited to the Software Citation WG." There is currently a lack of consensus on the use of RRIDs for software.

## Types of software

The principles and discussion in this document have generally been written to focus on software as source code. However, we recognize that some software is only available as an executable, a container, or a virtual machine image, while other software may be available as a service. We believe the principles apply to all of these forms of software, though the implementation of them will certainly differ based on software type. *When software is accessible as both source code and another type, we recommend that the source code be cited.*

## Access to software

The Accessibility principle (5) states that "software citations should permit and facilitate access to the software itself." This does not mean that the software must be freely available. Rather, the metadata should provide enough information that the software can be accessed. If the software is free, the metadata will likely provide an identifier that can be resolved to a URL pointing to the specific version of the software being cited. For commercial software, the metadata should still provide information on how to access the specific software, but this may be a company's product number or a link to a website that allows the software be purchased. As stated in the Persistence principle (4), we recognize that the software version may no longer be available, but it still should be cited along with information about how it was accessed.

## What an identifier should resolve to

While citing an identifier that points to, e.g., a GitHub repository can satisfy the principles of Unique Identification (3), Accessibility (5), and Specificity (6), such a repository cannot guarantee Persistence (4). *Therefore, we recommend that the software identifier should resolve to a persistent landing page that contains metadata and a link to the software itself, rather than directly to the source code files, repository, or executable.* This ensures

longevity of the software metadata—even perhaps beyond the lifespan of the software they describe. This is currently offered by services such as figshare and Zenodo (*GitHub, 2014*), which both generate persistent DataCite DOIs for submitted software. In addition, such landing pages can contain both human-readable metadata (e.g., the types shown by Table 2) as well as content-negotiable formats such as RDF or DOAP (https://github.com/edumbill/doap/).

### Updates to these principles

As this set of software citation principles has been created by the FORCE11 Software Citation Working Group (https://www.force11.org/group/software-citation-working-group), which will cease work and dissolve after publication of these principles, any updates will require a different FORCE11 working group to make them. As mentioned in section Future Work, we expect a follow-on working group to be established to promote the implementation of these principles, and it is possible that this group might find items that need correction or addition in these principles. *We recommend that this Software Citation Implementation Working Group be charged, in part, with updating these principles during its lifetime, and that FORCE11 should listen to community requests for later updates and respond by creating a new working group.*

## FUTURE WORK

Software citation principles without clear worked-through examples are of limited value to potential implementers, and so in addition to this principles document, the final deliverable of this working group will be an implementation paper outlining working examples for each of the use cases listed in section Use Cases.

Following these efforts, we expect that FORCE11 will start a new working group with the goals of supporting potential implementers of the software citation principles and concurrently developing potential metadata standards, loosely following the model of the FORCE11 Data Citation Working Group. Beyond the efforts of this new working group, additional effort should be focused on updating the overall academic credit/citation system.

## APPENDIX A

### Working group membership

Alberto Accomazzi, Harvard-Smithsonian CfA

Alice Allen, Astrophysics Source Code Library

Micah Altman, MIT

Jay Jay Billings, Oak Ridge National Laboratory

Carl Boettiger, University of California, Berkeley

Jed Brown, University of Colorado Boulder

Sou-Cheng T. Choi, NORC at the University of Chicago & Illinois Institute of Technology

Neil Chue Hong, Software Sustainability Institute

Tom Crick, Cardiff Metropolitan University

Mercè Crosas, IQSS, Harvard University

Scott Edmunds, GigaScience, BGI Hong Kong

Christopher Erdmann, Harvard-Smithsonian CfA

Martin Fenner, DataCite

Darel Finkbeiner, OSTI

Ian Gent, University of St Andrews, recomputation.org

Carole Goble, The University of Manchester, Software Sustainability Institute

Paul Groth, Elsevier Labs

Melissa Haendel, Oregon Health and Science University

Stephanie Hagstrom, FORCE11

Robert Hanisch, National Institute of Standards and Technology, One Degree Imager

Edwin Henneken, Harvard-Smithsonian CfA

Ivan Herman, World Wide Web Consortium (W3C)

James Howison, University of Texas

Lorraine Hwang, University of California, Davis

Thomas Ingraham, F1000Research

Matthew B. Jones, NCEAS, University of California, Santa Barbara

Catherine Jones, Science and Technology Facilities Council

Daniel S. Katz, University of Illinois (co-chair)

Alexander Konovalov, University of St Andrews

John Kratz, California Digital Library

Jennifer Lin, Public Library of Science

Frank Löffler, Louisiana State University

Brian Matthews, Science and Technology Facilities Council

Abigail Cabunoc Mayes, Mozilla Science Lab

Daniel Mietchen, National Institutes of Health

Bill Mills, TRIUMF

Evan Misshula, CUNY Graduate Center

August Muench, American Astronomical Society

Fiona Murphy, Independent Researcher

Lars Holm Nielsen, CERN

Kyle E. Niemeyer, Oregon State University (co-chair)

Karthik Ram, University of California, Berkeley

Fernando Rios, Johns Hopkins University

Ashley Sands, University of California, Los Angeles

Soren Scott, Independent Researcher

Frank J. Seinstra, Netherlands eScience Center

Arfon Smith, GitHub (co-chair)

Kaitlin Thaney, Mozilla Science Lab

Ilian Todorov, Science and Technology Facilities Council

Matt Turk, University of Illinois

Miguel de Val-Borro, Princeton University

Daan Van Hauwermeiren, Ghent University

Stijn Van Hoey, Ghent University

Belinda Weaver, The University of Queensland

Nic Weber, University of Washington iSchool

## APPENDIX B

### Software citation use cases

This appendix records an edited, extended description of the use cases discussed in section Use Cases, originally found in FORCE11 Software Citation Working Group. This discussion is not fully complete, and in some cases, it may not be fully self-consistent, but it is part of this paper as a record of one of the inputs to the principles. We expect that the follow-on Software Citation Implementation Group will further develop these use cases, including explaining in more detail how the software citation principles can be applied to each as part of working with the stakeholders to persuade them to actually implement the principles in their standard workflows.

***Researcher who uses someone else's software for a paper***

One of the most common use cases may be researchers who use someone else's software and want to cite it in a technical paper. This will be similar to existing practices for citing research artifacts in papers.

"Requirements" for researcher:

- Name of software
- Names of software authors/contributors
- Software version number and release date, or download date
- Location/repository, or contact name/email (if not publicly available)
- Citable DOI of software
- Format for citing software in text and in bibliography

Possible steps:

1. Software developers create CITATION file and associate with source code release/repository.
2. Researcher finds and uses software for research paper.
3. Researcher identifies citation metadata file (e.g., "CITATION" file) associated with downloaded/installed software source code or in online repository/published location.

CITATION file includes necessary citation metadata. CITATION file may include BibTeX entry, suggested citation format.

4. Researcher cites software appropriately, e.g., in methodology section; reference included in bibliography.

### Researcher who uses someone else's software for new software

In this case, a researcher develops new software that incorporates or depends on existing software. In order to credit the developer(s), the researcher will include citations in his/her source code, documentation, or other metadata in a similar manner to papers.

Requirements for researcher:

- Name of software
- Names of software authors/contributors
- Software version number and release date
- Location/repository
- Citable DOI of software
- Format for citing software in source code, documentation, or citation metadata file

Possible steps:

1. Assume that software developers have created a CITATION file and associated with the source code release/repository.

2. Researcher finds and uses software in the development of new software.

3. Researcher identifies citation metadata file (e.g., "CITATION" file) associated with downloaded/installed software source code or in online repository/published location. CITATION file includes necessary citation metadata. CITATION file may include BibTeX entry, suggested citation format.

4. Researcher cites software in source code, documentation, or other metadata-containing file.

### Researcher who contributes to someone else's software (open source project)

A researcher wants to contribute to someone else's software in the manner in which their contributions will be accepted and recognized.

Possible steps:

1. Researcher finds information about the software, and how contributors will be recognized

2. Researcher possibly submit a Contributor License Agreement (CLA) or Copyright Assignment Agreement (CAA) to allow the contributed content to be distributed with the software being contributed to

3. Researcher contributes to the software

4. Software maintainers accept contribution, recognize researcher's contribution, and update the software metadata as appropriate

### Researcher who wants to know who uses the researcher's software

This case is similar to a researcher who wants to find other papers/publications that cite a particular paper. A researcher wants to gauge the usage of her software within or across communities and measure its impact on research for both credit and funding.

Requirements:

- Uniquely identify software
- Indexed citations of software
- Indexed papers that use software

Steps:

1. Researcher finds software official name or unique DOI in metadata associated with downloaded/installed source code or in online repository/published location.
2. Researcher searches for software, may use online indexer (e.g., Scopus, Web of Science, Google Scholar) using software name or DOI.
3. Online indexer presents entry for software with list of citations, if any. Ideally, entry will also include metadata contained in software CITATION file and citation example.

### Researcher gets credit for software development at the academic/governmental institution, in professional career, etc

This case describes the need for a researcher who has contributed to software (by design, software engineering, development, testing, patching, documentation, training, evangelizing, etc.) to have their software work recognized by their employer or colleagues for the purpose of career advancement and increased professional reputation.

Requirements for researcher:

- Name of software
- Names of software authors/contributors
- Location/repository
- Citable DOI of software
- Format for citing software in an official CV, in a departmental/institutional review report, etc.
- Role in the software creation, that is linked to version or component
- Role in contributing to the software as a "package" (not just lines of code) development of benchmarks, testing, documentation, tutorials etc.

### Researcher who wants to "reproduce" another person/group's analysis

When a researcher wants to understand or verify a research results from another researcher, they would like to use the same software. Note that accessing the exact same software is necessary but not sufficient for reproducibility.

Requirements for researcher:

- Name of software
- Location/repository for the exact release that was used
- DOI or other persistent handle for that specific release
- Release has all components necessary for reproducing the work (Note: this ideally also means sample inputs and outputs)

### Researcher who wants to find a piece of software to implement a task

This is the case where a research is looking for software to use but wants to understand whether it is being used in a scholarly fashion. For example, a researcher searches through a software repository and finds a package that might be useful. They look to find whether it has been used by others in the scientific literature.

Requirements:

- Either the software documentation page has a reference to existing literature that makes use of it.
- There is a mechanism to look it up.

### Publisher wants to publish a software paper

This case asks what information regarding software is needed for a publisher who wants to publish a paper describing that software.

Requirements:

- Name of software
- Names of software authors/contributors
- Location/repository
- Citable DOI of software
- Format for citing software in JATS, for example, as well as references in the text itself

### Publisher who wants to publish papers that cite software

This case asks what information regarding software is needed for a publisher who wants to publish papers that cite that software.

Requirements for publisher:

- Name of software
- Names of software authors/contributors
- Location/repository
- Citable DOI of software
- Format for citing software in, e.g., JATS, as well as references in the text itself

### Indexer (e.g., Scopus, WoS, Scholar, MS Academic Search) who wants to build a catalog of software

Provide an index over the software that is used within the research domain. Track how that software is being used by different groups of researchers and to what ends.

  Requirements:

- Uniquely identify pieces of software used by the research literature
- Connect authors and organizations to that software
- Connect various software versions together

### Domain group (e.g., ASCL, bioCADDIE), Libraries, and Archives (e.g., University library, laboratory archive, etc.) wants to build a catalog/registry of institutional or domain software

There are two different examples here: One is building a catalog/archive of software produced by those affiliated with the institution. The other is along the lines of Sayeed Choudhury's note that "data are the new special collections." An institution may choose to build a catalog/archive of many things within a single topic or subject in order to secure all the software on a certain topic or build a collection that may draw users to their establishment, much like special collections now do for university libraries and archives.

### Repository showing scientific impact of holdings

A repository that archives and/or maintains a collection of software. The repository would like to address usage and impact of software in its holding. Usage would aid potential users whether the software is being actively maintained or developed or has been superseded. Both would help repository know how to direct resources, e.g., maintenance, training etc. This is similar to the case of a funder wanting to know the impact of funded work.

  Requirements:

- Code name, or a unique identifier
- Relationships to previous versions
- Connect to repository
- Connect to research

### Funder who wants to know how software they funded has been used

This use case is similar to "Repository showing scientific impact of holdings", where a funder wants to find out the use and impact and software that they supported. It is also similar to "Researcher who wants to know who uses the researcher's software."

### Evaluator or funder wants to evaluate contributions of a researcher

In this use case, an evaluator (e.g., academic administrator) or funder wants to evaluate the contributions of a researcher who develops software. This case is related to those

where researchers want to get credit for software development, or where organizations want to evaluate the impact of software itself.

### Reference management system used by researchers to author a manuscript

Reference management systems may need to be updated to internally understand that their is a software reference type, and to be able to output references to software in common formats.

Requirements for reference manager:

- Names of software authors/contributors
- Software version number and release date
- Location/repository
- Citable DOI of software or paper recommended for citation
- Format for citing software in citation metadata file
- Citation metadata tags embedded in DOI landing page/software project page for easy ingest

Possible steps:

1. Reference management system such as EndNote, Mendeley, Zotero, etc. builds affordances for software references.
2. Researcher finds software citation and adds it to their reference manager library, by (a) importing from the CITATION file (e.g., BibTeX, RIS), or (b) clicking on, e.g., an "add to Zotero library" widget in web browser.
3. Researcher writes a paper and uses the reference manager to generate citations or bibliography.

### Repository wants to publish mixed data/software packages

Domain and institutional data repositories have both data and software artifacts, and want to link these together in a provenance trace that can be cited. Sometimes the software is a separately identified artifact, but at other times software is included inside of data packages, and the researcher wants to cite the combined product.

## Use cases not adopted in the table
### Researcher who benchmarks someone else's software with or without modification on one or many hardware platforms for publication

This case describes the need for a researcher who has contributed to software (by design, software engineering, development, testing, patching, documentation, training, evangelizing, etc.) to have their software work recognized by their employer or colleagues for the purpose of career advancement and increased professional reputation.

Requirements for researcher:

- Name of software
- Names of software authors/contributors
- Software version number and release date
- Location/repository
- Citable DOI of software or paper recommended for citation
- Format for citing software in source code or citation metadata file

Possible steps:

1. Software developers create CITATION file and associate with source code release/repository.
2. Researcher finds and uses software in the development of new software.
3. Researcher identifies citation metadata file (e.g., CITATION file) associated with downloaded/installed software source code or in online repository/published location. CITATION file includes necessary citation metadata. CITATION file may include BibTeX entry, suggested citation format.
4. Researcher cites software in source code, documentation, or other metadata-containing file.

After review of this use case, we decided that based on the title this falls under use case 1, where a researcher uses someone else's software for a paper. Unlike use case 1, which is general in terms of the use of software, here the use leads to a benchmarking study—but the outcome in both cases is a paper that needs to cite the software.

### Researcher who wants to publish about a piece of software

The research wants to publish about a version of software they have produced. A key part of this use case is to be able to connect the given narrative to a specific version of the software in questions and connect that in large story.

Requirements:

- Name of software
- Names of software authors/contributors
- Location/repository
- Citable DOI of Software
- Links to older versions of software

This is similar to use case 1, other than the fact that the software developer(s) and paper author(s) will likely be the same person/people here.

### Researcher wants to record the software that generated some data

This is the case where a researcher is using some software to perform an analysis, either of a physical sample or of data. The researcher needs to know which version was used, for

example in case a bug was fixed. Note that knowing the software and its version is not sufficient to determine the "conditions" of the analysis, but they are essential.

Requirement: The analysis, or the generated data, has information about the software used.

This is also similar to use case 1, except in that case the research output is a paper, while here the output is a dataset.

### Researcher who wants to reproduce experience of use of a particular software implementation in context

Researcher is engaged in historical/cultural research, e.g., a study of video games as cultural artifacts.

Requirements:

- Name of software
- Software version number
- Documentation of the execution environment/context
- Location/repository for virtual machine (or equivalent) comprising both software and execution environment/context
- Persistent identifier associated with virtual machine instance (or equivalent) comprising both software and execution environment/context

Possible steps:

1. Researcher obtains persistent ID from citation
2. Research uses a persistent ID resolution service to resolve ID to a location of an executable VM instance in a repository
3. Researcher obtains VM in the repository, executes it, and interacts with software

This overlaps use case 6 (reproducing analysis), and so we decided not to include this as a distinct use case.

## APPENDIX C

### Feedback following FORCE2016

This appendix contains a record of comments made by the FORCE11 community on the draft Software Citation Principles, either directly via Hypothesis on the draft document (https://www.force11.org/softwarecitation-principles) posted following the FORCE2016 conference (https://www.force11.org/meetings/force2016) or via GitHub issues (https://github.com/force11/force11-scwg/issues), and the responses to these comments.

### On unique identification

I know this suggestion of a single unique identifier comes from the DOI perspective where it works pretty well, but I'm wondering if something different in the way of identification should be used for software. For creative works generally there is the FRBR model

(https://en.wikipedia.org/wiki/Functional_Requirements_for_Bibliographic_Records)
which defines several levels for a creative entity—"work," "expression," "manifestation,"
and "item." I think something along these lines are particularly relevant for software—it
is useful to be able to locate all uses of a particular piece of software no matter
what version (the "work" level—software identified by a particular name and purpose
over a period of time), but it is also important to specify the particular version used
in any given work ("expression"—the source code at the time of use) and in some
cases also the platform ("manifestation"—the compiled bytes including libraries,
for example a docker image). "Item" probably isn't relevant for software. That is, I
think a software citation perhaps could use THREE distinct unique identifiers, one
for the work itself, one for the specific version (source code), and possibly an additional
one for the actual downloadable binary image that can be run. Rather than leave it
implicit I think recognizing the different levels of citable record would be helpful
here. #F11SC

*Reply:* I interpret the requirement for "global uniqueness" as referring to the identifier
itself. Two different people can have the same name (not globally unique) but cannot
share a single ORCID (globally unique). Global uniqueness of the identifier does
not preclude multiple identifiers pointing to the same person. I think the suggestion
of differentiating between different software expressions/manifestations/items is a
reasonable one, but I don't think it relaxes the requirement for identifiers to be
globally unique.

**Our response:** We agree that there are valid points here, but on balance we don't feel
that the rewards from implementing this outweigh the practical challenges.

### On accessibility

Should this document address this in further detail? For example, "permit and facilitate
access" could be explored further. Should this be done through open access licensing?
repositories? Who's responsible for providing this access?

I am also wondering if this is a separate issue since "citing" traditionally pointed to
publications but did not necessarily address access. DOI, for example is stated, but doesn't
guarantee "access," so does this simply restating point 3, or should it provide
something new?

**Our response:** We agree that accessibility should receive further attention, which the
follow-on group focusing on implementation will provide. However, this is out of scope
for the document outlining the principles.

To the second point, accessibility provides information about access, but does not
guarantee access itself (e.g., paywalled article).

### On specificity

I am wondering if this should be folded into number 3 "Unique Identification." Both seem
to deal with the issue of identification and access.

**Our response:** A unique software identifier **can** point to the specific version/variant of
software, but it can also identify other things (collection of versions, repository, etc.),

while this principle deals with the need to identify the specific version of software used (via citation).

### On academic credit

A lot of software that were developed by non-academic engineers also contribute to academic research indirectly. Their names and contributions should also be credited. So removing "Academic" makes more sense?

*Reply:* This is a good point, though I think academic and non-academic credit are different, so perhaps we can add to this regarding non-academic credit, rather than removing "academic."

*Reply:* I agree with Daniel on this. Keep Academic and add non-academic.

**Our response:** We've made the bullet more general, just about credit, discussing academic credit and adding a sentence about non-academic credit as well.

### On citations in text

Although the focus here is on citations in the references, as a publisher, our experience is that most common practice of "citation" of data and software for authors is typically in the main body of the text. In order to encourage software to be treated and valued as a first-class research object, it is important that citations to it be positioned in the references as citations to articles and books are. However, it would be a missed opportunity if we did not leverage current practices of authors. This will also likely arise during implementation, as it has for the Data Citation Implementation Publisher Early Adopters Pilot. This could be addressed in future work on implementation.

**Our response:** In the principles, we propose that software should be cited in the references list, to recognize the primary role of software in research. However, this practice is not mutually exclusive with **also** referencing/citing software in the main body of a paper—as long as the software is cited in the references.

### On unique identification

Clearer instructions will be needed for authors on which version to cite. For BioMed Central journals, we ask authors to cite two versions of the software, an archived version (e.g., on Zenodo) as well as the current version (e.g., on GitHub). This is to ensure accessibility. However, if repositories and archives were to include a persistent link to the current version of the software, publishers could then instruct authors to cite only software with a UID, which wouldn't point to a current version, but would point to the version(s) used and would be a more accurate version of scientific record. Related to this point is the idea of group object identifiers. A need for group identifiers has been identified in the area of data (e.g., in the case of meta-analyses), and one could also identify a use case for these in the case of software, collecting metadata around all versions of a given software package. See blog here (https://blog.datacite.org/to-better-understand-research-communication-we-need-a-groid-group-object-identifier/).

**Our response:** We recommend citing the specific version of the software that was used. We expect that the unique identifier (e.g., DOI) will point to a landing page that directs to the repository/current version. However, this is more of a convenience issue that the software developers should address, rather than the author citing the software they used.

### On future work

For implementation we would recommend both consulting with adopters as well as developing metadata standards simultaneously rather than developing metadata standards and then pursuing early adopters implementation. The work early adopters are doing now for data citation will be able to be leveraged for software citation and the changes needed to do so could happen now. There is no need to wait on approval of new tagging for a specific metadata standard. Many publishers will have their own preferred metadata standards and so implementation could begin now with publishers, as long as we know what we want to capture. Future implementation groups might also consider levels of contribution. This is particularly relevant for software. Who is considered an author? For example, to what extent should authors of pull requests receive attribution? This might be considered in an FAQs group, or possibly an early adopters group.

**Our response:** We agree that metadata standards should be developed with the input of adopters, and have updated this text accordingly.

### Additional thoughts (not sure what section this applies to)

The principles do not address virtual machines. As these are becoming more common and relevant when addressing the reproducibility of research, it is important this "form" of software is acknowledged. The question remains in which cases should authors cite the current version, which the static archived version, and in which the virtual machine? In this way software is very much a unique evolving research object and might not fit perfectly into the same citation practices and structure as other research objects. In addition, software citation could possibly occur within the virtual machine. This could be added as a use case.

**Our response:** We feel this has been addressed in Section 5.8, with the explicit addition of virtual machines in addition to executables and containers. This is also an issue that should be addressed further by the follow-on implementation working group.

### On persistence of identifier vs. persistence of software

The persistence principle outlined in (4) is a key element in making software citeable. Where software has become part of the record of science not only the identifier and metadata of the software should be persistent, it should also be the goal to keep a persistent copy of the source code, where applicable. This links with the accessibility principle (5).

There are still many open questions about how to resolve package dependencies in the long term, therefore I would not make the persistent access to code a hard

requirement but may add something more specific towards preserving the record of science.

**Our response:** Our goal is for software citations to point to (persistent) archived source code, but we are not—nor could we—require this.

### Granularity of the citation

One of the key issues with any citation, whether document, individual, or software is the specificity of what is being cited. In the case of publications, there is almost zero specificity most of the time.

It's very easy to cite an entire package even though one function was used. Part of this problem is being solved in the Python world through this project (https://github.com/duecredit/duecredit).

Any citation should have the ability to specify more than just the obvious, but even the obvious would be a good starting point.

The citation/url should therefore allow for greater specificity within a code base. In general though, a provenance record of the workflow would be significantly more useful than a citation from a research perspective.

**Our response:** We agree that greater specificity is desirable in some cases, but we do not believe this rises to the level of what should be specified or discussed in the principles at this time.

### "Software citations should permit … access to the software itself"

Under the "Access" header, the data declaration states that:

Data citations should facilitate access to the data themselves.

Under the same header, the software declaration states:

Software citations should permit and facilitate access to the software itself.

The addition of "permit" suggests that software citations should also grant the user with permission to access the software. Is this intentional?

It doesn't seem like a good idea to make access a requirement for discovery, so "permit" might not be helpful in this sentence.

**Our response:** To avoid confusion, we removed "permit and" from the accessibility principle.

### Access to software: free vs commercial

The section talks about software that is "free" as well as "commercial" software. I am not sure whether this is about free as in freedom (or just gratis or freely available), since it is compared with commercial software, which is unrelated in general, see http://www.gnu.org/philosophy/words-to-avoid.html#Commercial.

I suppose that "free" should be replaced by "gratis" and "commercial" be replaced by "non-free" in that section.

**Our response:** We think this is sufficiently clear as written.

## ACKNOWLEDGEMENTS

While D. S. Katz prepared this material while employed at the NSF, any opinion, finding, and conclusions or recommendations expressed in this material are those of the authors and do not necessarily reflect the views of the NSF.

### Funding
Work by D. S. Katz was supported by the National Science Foundation (NSF) while working at the Foundation. Work by K. E. Niemeyer was supported in part by the NSF under grant ACI-1535065. The funders had no role in study design, data collection and analysis, decision to publish, or preparation of the manuscript.

### Grant Disclosures
The following grant information was disclosed by the authors:
NSF: ACI-1535065.

### Competing Interests
Arfon M. Smith is an employee of GitHub, Inc., San Francisco, California.

### Author Contributions
- Arfon M. Smith wrote the paper, prepared figures and/or tables, reviewed drafts of the paper.
- Daniel S. Katz wrote the paper, prepared figures and/or tables, reviewed drafts of the paper.
- Kyle E. Niemeyer wrote the paper, prepared figures and/or tables, reviewed drafts of the paper.

### Data Deposition
The research in this article did not generate, collect or analyse any raw data or code.

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
