# Peer review of "Software citation principles"

_PeerJ Computer Science, doi:10.7717/peerj-cs.86_

## Round 0.1 · original submission · Major Revisions

First of all, I would like to thank the reviewers, who have done a fantastic job and provided very insightful comments for improving the paper. I hope that the authors will take them into serious consideration in the revision.

From my personal perspective, this is a very good contribution, which addresses one of the most important topics in the "citation" domain, i.e. software citation. I totally agree with the reviewers about the readability of the paper, and I think this paper will deserve publication in PeerJ CS.

However, there are some aspects that should be clarified in the revision. In particular:

- I would like to see a new section that describes the presentation of these principles at FORCE 2016, and includes also a summary of the most important outcomes derived from the discussion the authors have done during the conference. Did such outcomes change some of the principles and/or other claims in this paper?

- The link between the principles and the requirements should be introduced explicitly, since it is not entirely clear how ones are connected to the others.

- A clear research question should be announced explicitly from the very beginning, and accompanied by a text that explains how the authors would like to address it.

- I suggest to include one or more appendixes with all the details about the use cases presented. This would make the paper more self-contained.

- The reviewers did not agree on some aspects described in the paper, e.g. "if the software authors ask that a paper should be cited, that should typically be respected". I would like to see an appropriate discussion on these points, so as to clearly address these disagreements.

All these aspects should be clearly addressed, revising the paper accordingly, so as to deserve a publication in PeerJ CS - that's why I've assigned "major revision" as final decision of the paper.

In addition, I would also like to see a full addressing of all the other relevant comments made by the reviewers, accompanied by an appropriate response letter.

·

Basic reporting

This paper argues for the adoption of software citations, basically for the same reasons that data citations have been advocated: Because software packages have become so important in science, they should be appropriately credited and referenced, which can be achieved if we extend the concept of citations in scientific articles.

The paper tackles an important problem and provides a very valuable foundation for further discussions and future implementations. In particular, the six software citation principles presented in the beginning are very valuable.

While I think this is a very useful paper, I see the following problems with the current version:

- The logical connections between the different parts of the paper are not clear. For example, how do the "basic requirements" of Table 2 map to the "principles" in Section 1? And are the principles presented in the beginning the *outcome* of this research (this is what lines 29+ seem to suggest) or are they *assumptions* of the approach to be discussed and evaluated (this is how they seem to be treated in Section 5).

- It is not clear whether the authors consider software citations to be just the first step towards a long-term vision, or whether they consider software citations the overall final solution. It seems that the first is the case, but the long-term vision is not discussed.

Experimental design

This paper doesn't have an experimental design in the strict sense, as it is more of a position paper than a research article. The PeerJ guidelines state that "the submission should clearly define the research question", which is currently not the case for this paper. However, I see a research question between the lines, which could be phrased as "how can citations be applied to software products to account for their increasing importance in scientific research?". By making such a research question explicit, the PeerJ requirements can be met.

The use cases described in Section 3 form the main method to answer the implicit research question. Given the importance of these use cases for this paper, I think reference [18] with the details about these use cases should be published in a more official way, either on a site like ArXiv or FigShare, or maybe even as supplemental material or appendix for this paper. I think having these use cases in the appendix of this paper would be the best solution, which would also increase the impact of these use cases and would strengthen the paper in general.

Validity of the findings

As noted above, the logical structure of the paper is unclear, and therefore it is not clear what exactly the main findings are. I think the main findings should be the principles stated in the beginning, and the use cases would then form the main method to arrive at these findings. Alternatively, one could label the principles as "assumptions" and apply the use case study to validate these assumptions. In any case, I think the findings are valid but they should be labeled as preliminary, as it is not possible to arrive at a final conclusion for such kinds of questions. Only the future can tell whether software citations really work out.

Additional comments

Below are some more specific comments.

line 88: "which will be presented ..." > "which was presented ..."

line 89: "We expect that this discussion may lead to a second, final version": So this paper is not the final version? Please clarify.

line 97: "the need for credit for application software underscores the need to overhaul the system of credit for all research products.": Very true and very important. But then maybe the concept of a "citation" is obsolete altogether, and we shouldn't be advocating "software citations", but something new altogether? Some more discussion on this would be valuable, I think. This relates to my comment above about a long-term vision.

line 238: "Citation is a record of software that is important to a research outcome": This is another important issue, but not much more is said about this. What makes a software "important" to a research outcome? Or, more generally, what is the semantics of a software citation?

line 254: "In addition, if the software authors ask that a paper should be cited, that should typically be respected": I think I disagree with that. Whether a paper should be cited or not should *not* depend on whether the given software package tells you to do so. Such pointers can be useful as suggestions, but not more than that in my opinion.

Lines 308+: You don't imply that this list is complete, but wouldn't an identifier of a software *branch* be another important type of identifier? For example, the Windows "version" of a software package. Also, I see in what situation one would use (1) or (2) with respect to software citations, but what is the purpose of (3) in the context of citations?

Line 325: "other software may be available as a service": This makes a huge difference, doesn't it? I think some more discussion on this would be helpful. For example, with software services it can be hard to find out what the official name of the software is, let alone the developers or version numbers. Also, does a software that is only available as a service (e.g. REST API) violate the Accessibility principle (5)?

I find it a bit weird to quote passages from the same paper like "The Accessibility principle (5) states that “....”" (line 329 and others).

In the discussion about Unique identification (lines 292+), I miss the discussion of hash values as version identifiers, as applied by Git for instance. As the first author is affiliated with GitHub, I assume this aspect was left out intentionally, but why? As basically all software is nowadays maintained with version control systems such as Git, these identifiers come for free and are by design unique. So, I would think they are an important aspect to discuss for software citations.

·

Basic reporting

[The submission must adhere to all PeerJ Computer Science policies]

This was not specifically checked by me, but I see no indication that it violates any reasonable policies.

[The article must be written in English ...]

Then paper is written in clear and respectful English. I found it easy to read.

[The article should include sufficient introduction and background...]

Apart from the principles themselves, I found the background provided to be very useful, relevant and well-referenced.

[The structure of the submitted article should conform to one of the templates...]

I believe the variation in structure from the norms suggested paper is well suited to its purpose. (See also my response at "Validity of findings".)

[Figures should be relevant to the content of the article...]

The two tables are relevant and helpful, though I did have a slight problem with the legibility of my printed copy of Table 2 (page 5). The distinction between solid and open circles was not immediately clear to me, and I might suggest using a slightly larger version of these symbols.

[The submission should be ‘self-contained,’ should represent an appropriate ‘unit of publication’...]

The submission appropriately addresses the topic indicated by its title ("principles"), and material deferred to separate publication ("technical solutions" and "examples") seems to be an entirely reasonable separation of concerns.

[All appropriate raw data has been made available in accordance with our Data Sharing policy]

N/A

[Formal results should include clear definitions of all terms and theorems, and detailed proofs]

N/A

Experimental design

N/A - see "Validity of findings"

Validity of the findings

The paper does not present a specific research finding, and as such does not feature a research question and experimental procedure that are expected of such papers. However, I think it makes in important contribution to the acceptance of software as part of the process and output of research activity, and reflects a consensus formed among a significant community of researchers. As such, I believe it justifies a place in the formally published literature.

The motivation for the work is clearly stated, and the paper adds clarity and depth to discussion of the role of software in research, and its attendant citation as such.

The process by which the proposed principles were arrived at is described across sections 1-4, and relevant information is referenced, so the principles articulated have a clear provenance and basis in research practice.

(I note that some research fields recognize "method papers", and I would judge this submission by the the standards of such papers rather than as a presentation of a specific research finding.)

Additional comments

The following comments are offered for the authors' consideration. I don't think any affect the fundamental value of the paper as presented.

Lines 32-35: as a justification for treating software differently from data, I found this passage raised more questions than answers, and that the comments here were uncompelling and unsupported. For example, some published data is updated on a frequency comparable with some software (e.g. the bi-monthly release schedule of FlyBase is comparable with many software projects - http://flybase.org/static_pages/docs/release_schedule.html). Also, I think the distinction between data and software is less clear-cut than the authors suggest (e.g. is a schema or stylesheet data or software?). I would see the difference between software and data as more akin to the difference between research methods and research findings, but that is just a personal opinion. My suggestion would be that this justification is not really necessary to the overall purpose of the paper, and might be dropped without significant loss of value.

Line 57, et seq: [Nit] I would find this easier to read if the sentence introducing the list of reasons were moved to the same page as the list itself.

Line 88: "will be presented" refers to an event now in the past: change to "was presented"?

Line 113: [Nit] "work using software" appears to be very broadly scoped, and could apply to just about any activity using a computer. Suggest "research using software".

Table 2, page 5: I had a problem with the legibility of my printed copy of this table. The distinction between solid and open circles was not immediately clear to me. I might suggest using a slightly larger version of these symbols, or maybe more distinct symbol shapes.

Table 2, page 5: [Comment] This table contains a lot of information, and much food for debate. I was surprised by some ommissions, which may mean that I wasn't fully understanding the concepts being presented:

(a) I find it hard to imagine any of the use cases not benefiting from availability of a "Description", especially 1, 2, 9, and 16.
(b) I would have expected "Indexed citations" to be useful for 5: getting credit.
(c) For publishing a software paper, I would expect all of the basic requirements other than "Indexed citations" to be at least beneficial.
(d) For 12: build a software catalog/registry, would there not be a role for indexed citations?
(e) For 16: Store software try, I'd expect pretty much all of the Basic requirements other than "Indexed citations" to be relevant, and particularly "Description" and "Keywords".
(f) I think there is one use-case missing, which I think is commonly faced by researchers: viz. "Evaluate suitability of software for a given task" - I think this is distinct from discovery, and typically involves making a choice between different available software packages, or to use an existing package, modify an existing package or develop something new.

Line 175: [Nit] "in(1)" Missing space?

Lines 241-242: "Similarly, the software metadata that is recorded as part of data provenance should be a superset of the metadata recorded as part of software citation." I found this non sequitur (or am not seeing the logical connection). I can imagine reasons to cite software that does not form part of the provenance chain of results presented (e.g. alternatives considered).

Lines 245-247: "In general, we intend the software citation principles to cover the minimum of what is necessary for software citation for the purpose of software identification. Other use cases (e.g., provenance, reproducibility) may lead to additional requirements (i.e., enhanced metadata)." This statement feels to me to be at odds with table 2, which presents "Use cases and basic metadata requirements for software citation", which includes "reproducibility" as a use-case. This leaves me wondering if the use cases in table 2 are going beyond what is strictly needed for the purposes of software citation.

Line 327-328: "When software exists as both source code and another type, we recommend that the source code be cited". The term "exists" here doesn't seem quite right to me. E.g., proprietary software may "exist" as source code, but be unavailable. Suggest "exists and is accessible" or just "is accessible"?

Line 348: "Updates to this document". This paper is being presented for consideration as a part of the formal academic record - as such, I don't think "this document" is something which may be updated. Suggest "Updates to these principles".

---

## Round 0.2 · Minor Revisions

First of all, I would like to thank the authors for having addressed all my comments and reviewers' suggestions in their revision (and clearly explained their modifications in their response letter), and thanks again to the reviewers for their wonderful job.

While the paper is now in a good shape for being published, there are just minor issues that should be properly addressed before having it accepted for the final publication. Such issues are highlighted by the reviewer, and concern the new parts the authors have added, in particular:

1. the comparison between software and data;
2. the assignment of an identifier for explicit versions of a software by means of repository capabilities (e.g. hashes).

This is not a huge amount of work to add, as far as I can see, but a clear discussion (even supported by appropriate references) would be beneficial for the article.

Thanks again for your wonderful work.

·

Basic reporting

The authors followed most of my suggestions, and where they did not, they mostly
presented convincing arguments. I like in particular that there is now an
appendix with the use cases.

However, there are two aspects of the latest version that I find not very
convincing:

1. The listed differences between software and data (lines 100-114) are not
convincing in my opinion:

- "Software can be used to express or explain concepts, unlike data": I don't
understand this.

- "Software is updated more frequently that papers or data" (typo: "that"
instead of "than"): This was questioned also by the second reviewer, and I
don't think the issue is resolved. The footnote is unsupported and seems to
partly contradict the main point. (I can imagine that maybe only *popular*
software tends to be updated frequently, whereas *popular* dataset are often
not.)

- "Software suffers from a different type of bit rot than data": I am not
convinced that this is really a different *type* of bit rot. In particular
datasets in proprietary formats seem to suffer from exactly the same problems.

- "Software is frequently built to use other software": True, though similar
things can occur in datasets too, in particular with respect to Linked Data,
where datasets typically use a number of third-party ontologies and
vocabularies.

- "Software is generally smaller than data": Another unsupported claim, and I am
not fully convinced that this is true. There must be a large number of very
small spreadsheets of data. So, on average data might be smaller than
software. The difference is certainly true on the upper end of the scale
though: the largest datasets are larger than the largest pieces of software.

- "Software teams can be large and multidisciplinary, ...": I think this applies
to data too.

- "The lifetime of software is generally not as long as that of data.": Again
unsupported, and again I can imagine this to be false. Many datasets, in
particular small ones, might be stillbirths (i.e. they are never used). The
same might be true for software, but possibly to a lesser degree. And software
might commonly evolve, whereas data might more often be replaced by a
different and newer dataset, in which case evolving software would live
longer.

2. I am also not convinced by the arguments against hashes as identifiers:

- "commit references are not guaranteed to be permanent": identifiers consisting
of strong hash values *are* guaranteed to be permanent in the sense that they
can forever only represent what they were meant to represent, i.e. the bits
and bytes on which the hash was calculated (which, in the case of Git, is the
entire version history).

- "A repository address and version number does not guarantee that the software
is available at a particular (resolvable) URL". It depends. A URL like
https://github.com/torvalds/linux/commit/a0cba2179ea4c1820fce2ee046b6ed90ecc56196
*does* come with a hash and can be resolved to get to the actual software.
There is no guarantee that github.com will be available forever, but this is
also true for publisher sites and therefore DOIs.

- "A particular version number/commit reference may not represent a “preferred”
point at which to cite the software from the perspective of the package
authors.": Hashes certainly don't prevent you from defining preferred versions
for future reference. But they come with the *possibility* to identify all
versions, which I would in fact consider an important feature. If I want to
make my research perfectly reproducible, I should be able to cite the exact
version that I used, whether that happens to be a preferred version or not.

- "When we talk about a “Unique identifier” in the document, we are referring to
unique, persistent, and machine-actionable identifiers such as a DOI, ARK, or
PURL; we do not believe a repository address with a version number is
sufficient." [from the rebuttal letter]: Identifiers that include hashes are
in important ways *more* unique, *more* persistent, and *more*
machine-actionable than DOIs or PURLs. Strong hashing algorithms guarantee
uniqueness (no other input can be constructed for the same hash), are in
practice probably more persistent (for example, there exist independent
complete copies of all GitHub content, so even if github.com closes for good,
the cited software could still be found), and can be automatically verified
against their supposed content. (The above holds for *strong* hash functions,
and Git's SHA-1 is no longer considered strong - even though not a single
collision has been found until now - but this doesn't invalidate the general
points.)

Experimental design

No further comments.

Validity of the findings

No further comments.

---

## Author Rebuttal · Round 0.2

**School of Mechanical, Industrial, and Manufacturing Engineering**
Oregon State University, 320 Rogers Hall, Corvallis, Oregon 97331
541-737-5614 | http://kyleniemeyer.com | kyle.niemeyer@oregonstate.edu

July 30, 2016

Dear *PeerJ CS* Editors,

We thank the reviewers and editor for taking the time to thoughtfully review our paper, and for their helpful comments. We have addressed all of the comments and suggested changes here.

We believe that the manuscript is now suitable for publication in *PeerJ CS*, and hope you agree.

Sincerely yours,

Kyle E. Niemeyer, PhD
Assistant Professor, School of Mechanical, Industrial, and Manufacturing Engineering
Oregon State University

(on behalf of the authors)

We thank the editor and reviewers for their helpful comments, and have addressed their requests and questions here. Specific changes or additions can be found in the marked revision.

# Editor:

*First of all, I would like to thank the reviewers, who have done a fantastic job and provided very insightful comments for improving the paper. I hope that the authors will take them into serious consideration in the revision.*

*From my personal perspective, this is a very good contribution, which addresses one of the most important topics in the "citation" domain, i.e. software citation. I totally agree with the reviewers about the readability of the paper, and I think this paper will deserve publication in PeerJ CS.*

*However, there are some aspects that should be clarified in the revision. In particular:*

1. **Comment:** *I would like to see a new section that describes the presentation of these principles at FORCE 2016, and includes also a summary of the most important outcomes derived from the discussion the authors have done during the conference. Did such outcomes change some of the principles and/or other claims in this paper?*

   **Our response:** We have added some additional description of these activities in Section 2, and also a new appendix that records the feedback and discussion following the FORCE2016 workshop and presentation. The workshop and feedback following presentation to the broader FORCE11 community led to some edits to the use cases and discussion, but not the principles themselves.

2. **Comment:** *The link between the principles and the requirements should be introduced explicitly, since it is not entirely clear how ones are connected to the others.*

   **Our response:** We have made some changes that we hope improve the logical flow of the document, and also added a new section (3) that explicitly describes the process by which we developed the principles. This includes the role that the use cases played in this process.

3. **Comment:** *A clear research question should be announced explicitly from the very beginning, and accompanied by a text that explains how the authors would like to address it.*

   **Our response:** We agree with Reviewer 2 on this issue, as this represents more of a position paper than an experimental paper. That said, we would be willing to add an explicit research question and discuss the "experimental" design if required, but feel this would be an artificial construct that doesn't add much to the paper.

4. **Comment:** *I suggest to include one or more appendixes with all the details about the use cases presented. This would make the paper more self-contained.*

   **Our response:** We agree that adding the external use case material would help make the paper more complete, and have added it as a new Appendix B. In addition, we added the public/FORCE11 community feedback (and our responses) as a new Appendix C.

5. **Comment:** *The reviewers did not agree on some aspects described in the paper, e.g. "if the software authors ask that a paper should be cited, that should typically be respected". I would like to see an appropriate discussion on these points, so as to clearly address these disagreements.*

   **Our response:** We have either made appropriate changes or discussed these points.

# Reviewer #1 (Tobias Kuhn):

*This paper argues for the adoption of software citations, basically for the same reasons that data citations have been advocated: Because software packages have become so important in science, they should be appropriately credited and referenced, which can be achieved if we extend the concept of citations in scientific articles.*

*The paper tackles an important problem and provides a very valuable foundation for further discussions and future implementations. In particular, the six software citation principles presented in the beginning are very valuable.*

*While I think this is a very useful paper, I see the following problems with the current version:*

1. **Reviewer comment:** *The logical connections between the different parts of the paper are not clear. For example, how do the "basic requirements" of Table 2 map to the "principles" in Section 1? And are the principles presented in the beginning the \*outcome\* of this research (this is what lines 29+ seem to suggest) or are they \*assumptions\* of the approach to be discussed and evaluated (this is how they seem to be treated in Section 5).*

   **Our response:** We view the principles presented at the beginning as the main outcomes/results of this work, with the process of producing them (discussed in the new Section 3) based on a synthesis of information from the use cases and related work. Then, Section 6 (previously 5) discusses the "results". Our modifications to the structure of the paper, and changes that we hope improve the logical flow, should help make these connections more clear.

2. **Reviewer comment:** *It is not clear whether the authors consider software citations to be just the first step towards a long-term vision, or whether they consider software citations the overall final solution. It seems that the first is the case, but the long-term vision is not discussed.*

   **Our response:** Ultimately we believe that software citation is just a part of a more complete solution for the sharing and citation of all products of scholarly research. We have updated the language in the first paragraph of §2 to reflect this.

3. **Reviewer comment:** *This paper doesn't have an experimental design in the strict sense, as it is more of a position paper than a research article. The PeerJ guidelines state that "the submission should clearly define the research question", which is currently not the case for this paper. However, I see a research question between the lines, which could be phrased as "how can citations be applied to software products to account for their increasing importance in scientific research?". By making such a research question explicit, the PeerJ requirements can be met.*

   **Our response:** We agree that this paper is more of a position paper than research article, and thus that creating a research question would be an artificial construct. We are willing to make this change if required, but feel it wouldn't add much to the paper.

   We modeled our process on the work done by the FORCE2016 Data Citation group, which included publishing a *PeerJ CS* paper (https://peerj.com/articles/cs-1/) which was written in a similar manner to our paper.

4. **Reviewer comment:** *The use cases described in Section 3 form the main method to answer the implicit research question. Given the importance of these use cases for this paper, I think reference [18] with the details about these use cases should be published in a more official way, either on a site like ArXiv or FigShare, or maybe even as supplemental material or appendix for this paper. I think having these use cases in the appendix of this paper would be the best solution, which would also increase the impact of these use cases and would strengthen the paper in general.*

   **Our response:** We agree, and have added the full use cases text (with some edits for clarity and completion) as the new Appendix B for completeness. We also added the comments (with our responses) made publicly following the presentation of the draft principles document at the FORCE2016 conference, as Appendix C.

5. **Reviewer comment:** *As noted above, the logical structure of the paper is unclear, and therefore it is not clear what exactly the main findings are. I think the main findings should be the principles stated in the beginning, and the use cases would then form the main method to arrive at these findings. Alternatively, one could label the principles as "assumptions" and apply the use case study to validate these assumptions. In any case, I think the findings are valid but they should be labeled as preliminary, as it is not possible to arrive at a final conclusion for such kinds of questions. Only the future can tell whether software citations really work out.*

   **Our response:** We agree that the main findings of the paper are the software citation principles, and as described in our response to point 1 above, we have updated the introduction and made some structural changes to make this clear.

   However, we respectfully don't agree that the principles are preliminary, but would describe them as reflecting the group's current knowledge and experience. We agree that "only the future can tell whether software citations really work out," but this is true of most papers: they capture the current state of scientific knowledge in an area.

6. **Reviewer comment:** *line 88: "which will be presented ..." to "which was presented ..."*

   **Our response:** We agree, and have made this change.

7. **Reviewer comment:** *line 89: "We expect that this discussion may lead to a second, final version": So this paper is not the final version? Please clarify.*

   **Our response:** Thank you for pointing this error out. We have corrected this statement to indicate that the past discussion led to a second version (what was submitted).

8. **Reviewer comment:** *line 97: "the need for credit for application software underscores the need to overhaul the system of credit for all research products.": Very true and very important. But then maybe the concept of a "citation" is obsolete altogether, and we shouldn't be advocating "software citations", but something new altogether? Some more discussion on this would be valuable, I think. This relates to my comment above about a long-term vision.*

   **Our response:** We have added new language to the Motivation section (§2) describing what a more complete solution might look like, and also included an additional sentence and citation to existing work on this topic towards the end of §3.

9. **Reviewer comment:** *line 238: "Citation is a record of software that is important to a research outcome": This is another important issue, but not much more is said about this. What makes a software "important" to a research outcome? Or, more generally, what is the semantics of a software citation?*

   **Our response:** We have revised this discussion to better address these issues. In particular, we added some references that discuss the importance of specific software versions to research results (including data). For a particular study, it is up the authors to determine which software is important, and peer-reviewers to decide if the authors have done this sufficiently well, according to community standards, similarly to how citation of papers in a paper is decided and judged.

10. **Reviewer comment:** *line 254: "In addition, if the software authors ask that a paper should be cited, that should typically be respected": I think I disagree with that. Whether a paper should be cited or not should \*not\* depend on whether the given software package tells you to do so. Such pointers can be useful as suggestions, but not more than that in my opinion.*

    **Our response:** We have modified the text to make it clear that this is a recommendation from the working group; the statement reflects an agreed position of the entire working group. The modification should make it clear that this is a recommendation only.

11. **Reviewer comment:** *Lines 308+: You don't imply that this list is complete, but wouldn't an identifier of a software \*branch\* be another important type of identifier? For example, the Windows "version" of a software package. Also, I see in what situation one would use (1) or (2) with respect to software citations, but what is the purpose of (3) in the context of citations?*

**Our response:** Principle (6) states that "Software identification should be as specific as necessary", and we consider that the specific version includes the branch.

We agree that (3) is not relevant in the context of citation, and have removed it from the discussion. However, it is a valid identifier that could be used in other contexts.

12. **Reviewer comment:** *Line 325: "other software may be available as a service": This makes a huge difference, doesn't it? I think some more discussion on this would be helpful. For example, with software services it can be hard to find out what the official name of the software is, let alone the developers or version numbers. Also, does a software that is only available as a service (e.g. REST API) violate the Accessibility principle (5)?*

**Our response:** Software as a service is a fact, and we need a method to cite it—even though some metadata may not exist or be available, we need to do the best we can. The same issues would hold for commercial software or that available only as an executable. If you can't find the name of software, should be able to find the name of the service/endpoint or function being used.

Regarding accessibility, ideally we would like the source to be available, but if not we still need a way to cite. In this case, the Accessibility principle would apply to the service itself, rather than the source.

13. **Reviewer comment:** *I find it a bit weird to quote passages from the same paper like "The Accessibility principle (5) states that ?....?" (line 329 and others).*

**Our response:** We feel it is more clear to a reader to quote the basis by which we are making follow-on comments, rather than just pointing back to a previous part of the paper.

14. **Reviewer comment:** *In the discussion about Unique identification (lines 292+), I miss the discussion of hash values as version identifiers, as applied by Git for instance. As the first author is affiliated with GitHub, I assume this aspect was left out intentionally, but why? As basically all software is nowadays maintained with version control systems such as Git, these identifiers come for free and are by design unique. So, I would think they are an important aspect to discuss for software citations.*

**Our response:** While we agree there is value in including the explicit version (e.g., Git SHA1 hash, Subversion revision number) of the software in a software citation, we also believe there are a number of reasons that this, by itself, is not sufficient for the purposes of software citation:

- Version numbers are not guaranteed to be permanent: projects can be migrated to new version control systems (e.g., SVN to Git). In addition, it is possible to overwrite versions (e.g., Git force-pushing).
- Stating a version number doesn't guarantee that the software is available at a particular (web-resolvable) URL.
- When we talk about a "Unique identifier" in the document, we are referring to unique, persistent, and machine-actionable identifiers such as a DOI, ARK, or PURL; we do not believe a repository address with a version number is sufficient.

We recommend that software citations capture the versions of the code expressed as (ideally) a semantic version (SemVer) number (http://semver.org/). These are points in time at which the authors of the code made the explicit decision to release the code. We feel a SemVer release number together with a DOI (and associated archive in, e.g., Zenodo) is the most robust and future-proof solution. That said, in the absence of "recommended" identifiers (such as a DOI) a Git SHA together with a link to a Git repository would be preferable to not having this information.

We would also like to point out that there will be a follow-on working group that will produce recommended implementations for software citations in a particular setting. Ultimately, the decision to (optionally) include something like Git SHAs is left for that implementation group to consider.

We have updated the paper text to reflect some of these ideas.

# Reviewer #2 (Graham Klyne):

1. **Reviewer comment:** *The two tables are relevant and helpful, though I did have a slight problem with the legibility of my printed copy of Table 2 (page 5). The distinction between solid and open circles was not immediately clear to me, and I might suggest using a slightly larger version of these symbols.*

   **Our response:** We agree, and have changed the open symbol to a plus sign to improve legibility.

2. **Reviewer comment:** *The paper does not present a specific research finding, and as such does not feature a research question and experimental procedure that are expected of such papers. However, I think it makes in important contribution to the acceptance of software as part of the process and output of research activity, and reflects a consensus formed among a significant community of researchers. As such, I believe it justifies a place in the formally published literature.*

   **Our response:** We agree with the reviewer's description of our paper, and appreciate the view that it should be formally published.

3. **Reviewer comment:** *Lines 32–35: as a justification for treating software differently from data, I found this passage raised more questions than answers, and that the comments here were uncompelling and unsupported. For example, some published data is updated on a frequency comparable with some software (e.g. the bi-monthly release schedule of FlyBase is comparable with many software projects - http:// flybase. org/ static_ pages/ docs/ release_ schedule. html ). Also, I think the distinction between data and software is less clear-cut than the authors suggest (e.g. is a schema or stylesheet data or software?). I would see the difference between software and data as more akin to the difference between research methods and research findings, but that is just a personal opinion. My suggestion would be that this justification is not really necessary to the overall purpose of the paper, and might be dropped without significant loss of value.*

   **Our response:** We do agree with the reviewer's specific point about release frequency, and have added a note about this. We have also expanded the set of differences to make the case more convincing.

   However, we respectfully disagree with the reviewer about the value of this section. Since our work on software citation is being done following previous work on data citation (including the *PeerJ CS* paper on data citation (https://peerj.com/articles/cs-1/), and since our software citation principles have some difference from the data citation principles, we feel it is important to explain the high level reasons why differences might be expected.

4. **Reviewer comment:** *Line 57, et seq: [Nit] I would find this easier to read if the sentence introducing the list of reasons were moved to the same page as the list itself.*

   **Our response:** We agree that this would be easier to read, but believe the formatting will change in the final published paper.

5. **Reviewer comment:** *Line 88: "will be presented" refers to an event now in the past: change to "was presented"?*

   **Our response:** We agree, and have made this change.

6. **Reviewer comment:** *Line 113: [Nit] "work using software" appears to be very broadly scoped, and could apply to just about any activity using a computer. Suggest "research using software".*

   **Our response:** We agree, and have made this change.

7. **Reviewer comment:** *Table 2, page 5: I had a problem with the legibility of my printed copy of this table. The distinction between solid and open circles was not immediately clear to me. I might suggest using a slightly larger version of these symbols, or maybe more distinct symbol shapes.*

   **Our response:** We agree, and have replaced the open bullet with a plus sign for better legibility.

8. **Reviewer comment:** *Table 2, page 5: [Comment] This table contains a lot of information, and much food for debate. I was surprised by some omissions, which may mean that I wasn't fully understanding the concepts being presented:*

(a) *I find it hard to imagine any of the use cases not benefiting from availability of a "Description", especially 1, 2, 9, and 16.*

(b) *I would have expected "Indexed citations" to be useful for 5: getting credit.*

(c) *For publishing a software paper, I would expect all of the basic requirements other than "Indexed citations" to be at least beneficial.*

(d) *For 12: build a software catalog/registry, would there not be a role for indexed citations?*

(e) *For 16: Store software entry, I'd expect pretty much all of the Basic requirements other than "Indexed citations" to be relevant, and particularly "Description" and "Keywords".*

(f) *I think there is one use-case missing, which I think is commonly faced by researchers: viz. "Evaluate suitability of software for a given task" - I think this is distinct from discovery, and typically involves making a choice between different available software packages, or to use an existing package, modify an existing package or develop something new.*

**Our response:**

(a) We agree for use cases 1 and 2, and have added "Description" as beneficial metadata. However, we feel that a description would not really be necessary for use case 9 from the perspective of a publisher, and that a citation manager would not really need or benefit much from the description.

(b) We agree that use case 5 could benefit from indexed citations, and added that as beneficial metadata.

(c) We feel that the metadata omitted would not be necessary from the perspective of the publisher, the stakeholder for this use case.

(d) We feel that indexed citations are not particularly relevant for a catalog or registry (based on input from administrators of such entities), unless they want to display citations of the software—but this would be use case 4. Thus, we prefer to leave this as is.

(e) We agree that indexed citations are not necessary for use case 16, and have removed this. We also agree that the use case could benefit from description and keywords, in the same way that citation managers likely contain an abstract and keywords for papers.

(f) We intended "find" to mean more than discovery, but since the appendix was not present in the first submission this was not clear. We have altered the use case to be "Find/choose software to implement task" to make it clear that this use case goes beyond discovery.

Also, we have added the full use case discussion material as a new Appendix B; some of that content may be incomplete or wrong, but is included as a record to support the process that lead to the use cases material in the document body. The use cases will also be subject to further work by the follow-on implementation group.

9. **Reviewer comment:** *Line 175: [Nit] "in(1)" Missing space?*

   **Our response:** This has been fixed.

10. **Reviewer comment:** *Lines 241–242: "Similarly, the software metadata that is recorded as part of data provenance should be a superset of the metadata recorded as part of software citation." I found this non sequitur (or am not seeing the logical connection). I can imagine reasons to cite software that does not form part of the provenance chain of results presented (e.g. alternatives considered).*

    **Our response:** We agree with the reviewer. We had been thinking of citation only in the context of software that had been used, not the other uses of citation. We have changed the text to acknowledge these other uses of citation.

11. **Reviewer comment:** *Lines 245–247: "In general, we intend the software citation principles to cover the minimum of what is necessary for software citation for the purpose of software identification. Other use cases (e.g., provenance, reproducibility) may lead to additional requirements (i.e., enhanced metadata)." This statement feels to me to be at odds with table 2, which presents "Use cases and basic*

*metadata requirements for software citation", which includes "reproducibility" as a use-case. This leaves me wondering if the use cases in table 2 are going beyond what is strictly needed for the purposes of software citation.*

**Our response:** We have rephrased this statement to clarify our meaning.

The requirements given in the use cases table only indicate the metadata needed for citation of software for each use case, and not necessarily all the requirements needed to actually complete each use case. For example, use case 7 "Benchmark software" does not include requirements for performing actual benchmarking, including hardware, performance monitoring software, performance modeling knowledge, etc.

12. **Reviewer comment:** *Line 327–328: "When software exists as both source code and another type, we recommend that the source code be cited". The term "exists" here doesn't seem quite right to me. E.g., proprietary software may "exist" as source code, but be unavailable. Suggest "exists and is accessible" or just "is accessible"?*

    **Our response:** Agreed; this change has been made.

13. **Reviewer comment:** *Line 348: "Updates to this document". This paper is being presented for consideration as a part of the formal academic record - as such, I don't think "this document" is something which may be updated. Suggest "Updates to these principles".*

    **Our response:** Agreed; this change has been made.

---

## Round 0.3 · accepted · Accept

I would like to thank the authors for their revisions and responses, and the reviewers for their incredible contribution to this paper. I think they have done an impressive job to finalise this work as it is in the current form.

Thus, I'm pleased to say that it is now ready to be published as part of PeerJ Computer Science.

Thanks again for your incredible work!